# The Effectiveness of Zinc-Biofortified Wheat Flour Intake on the Growth and Morbidity Outcomes of Rural Pakistani Children and Adolescent Girls: A Cluster-Randomised, Double-Blind, Controlled Trial

**DOI:** 10.3390/nu17071137

**Published:** 2025-03-25

**Authors:** Swarnim Gupta, Mukhtiar Zaman, Sadia Fatima, Victoria H. Moran, Jonathan K. Sinclair, Nicola M. Lowe

**Affiliations:** 1Centre for Global Development, University of Central Lancashire, Preston PR1 2HE, UK; sgupta6@uclan.ac.uk (S.G.); vlmoran@uclan.ac.uk (V.H.M.); jksinclair@uclan.ac.uk (J.K.S.); 2Department of Pulmonology, Rehman Medical Institute, Peshawar 25000, Pakistan; mukhtiar.zaman@rmi.edu.pk; 3Institute of Basic Medical Sciences, Khyber Medical University, Peshawar 25100, Pakistan; sadiafatima@kmu.edu.pk

**Keywords:** zinc, biofortification, adolescent girls, children, growth, morbidity, wheat flour, Pakistan, effectiveness

## Abstract

Background: Zinc-biofortified cereals are a promising strategy to combat zinc deficiency, though evidence on health outcomes is limited. This study assessed the effectiveness of consuming zinc-biofortified wheat flour on growth and zinc-related morbidity among adolescent girls (10–16 years; N = 517) and children (1–5 years; N = 517) living in rural north-west Pakistan. Methods: In this double-blind, cluster-randomised controlled effectiveness trial, 486 households received either zinc-biofortified or control wheat flour for 25 weeks. Anthropometric measurements and lung function tests (LFTs) were performed at the beginning, middle, and endline. Data on the incidence and duration of respiratory tract infection (RTI) and diarrhoea in the preceding two weeks were collected fortnightly. Analyses included baseline-adjusted linear mixed models for continuous outcomes and Pearson’s chi-square for categorical data. Results: At a zinc differential of 3.7 mg/kg for adolescent girls provided by zinc-biofortified wheat flour, the intervention had no significant effect on height or weight. For children, head circumference was significantly greater in the biofortified group at endline (control 48.47 ± 2.03 cm vs. intervention 48.76 ± 1.82 cm; *p* = 0.003), with no differences in other anthropometric parameters. Towards the end of the trial, a lower incidence of RTIs was reported in the intervention arm compared to the control arm for both children (week 26: control 27.4% vs. intervention 17.6%, *p* = 0.036) and adolescent girls (week 24: control 19.3% vs. intervention 11.5%, *p* = 0.037; week 26: control 14.5% vs. intervention 6.1%, *p* = 0.014). When the longitudinal prevalence (cumulative days of sickness as a percentage of total days) of RTI was considered, no treatment effects were observed. No benefits of treatment were reported for diarrhoea or LFT. Conclusions: The provision of zinc-biofortified wheat flour for 25 weeks did not have a significant effect on the growth of adolescent girls but modestly improved head circumference in children. Longer-term interventions are needed to monitor changes in functional outcomes with the national scale-up of zinc-biofortified wheat varieties.

## 1. Introduction

Micronutrient deficiencies, also referred to as “hidden hunger”, is a significant and persistent global public health concern [1,2]. Zinc is an indispensable micronutrient for human health; however, 17.3% of the global population has inadequate zinc intake, with rates as high as 30% in South Asia, where most countries are classified as low- and middle-income countries (LMICs) [3]. Recent estimates show that the prevalence of zinc deficiency among young children and non-pregnant women of reproductive age (WRA) exceeds 20% for most LMICs, indicating a need for public health interventions [4]. Zinc deficiency occurs more frequently in LMICs due to the consumption of foods low in zinc and/or its bioavailability, as plant-based diets are often rich in inhibitors of zinc absorption, particularly phytate—the most potent known inhibitor, which forms an insoluble complex with zinc in the small intestine, rendering it unavailable for absorption [5,6]. This issue is further compounded by infectious diseases leading to a vicious cycle that undermines human development and limits overall potential. Zinc deficiency impairs growth, immune function, reproduction, and neurodevelopment, with the greatest impact in low-resource settings, where it is linked to high rates of child stunting, increased morbidity and mortality, and poor maternal health outcomes [7]. Zinc is associated with respiratory health and diarrhoea in vulnerable populations [8,9,10]. In LMICs, zinc deficiency accounts for up to 4.4% of childhood deaths and 1.2% of the overall disease burden for children under 5 years of age, with a higher (3.8%) disease burden among children aged 6 months to 5 years [11]. Specifically, zinc deficiency is estimated to contribute to 14.4% of all diarrhoea-related deaths and 6.7% of all pneumonia-related deaths in children under 5 years [11].

In Pakistan, zinc deficiency affects 22.1% of WRA and 18.6% of children under five years, with regional and rural–urban disparities [12]. Rural areas experience higher prevalence, particularly in marginalised communities where poverty, low education, poor infrastructure, and limited healthcare access exacerbate the issue. We have reported that 30% of WRA (22–48 years) [13] and almost 70% of adolescent girls (10–16 years) [14] in marginalised communities in the Khyber Pakhtunkhwa province are zinc deficient.

Agricultural interventions, such as biofortification, that enhance the nutritive value of crops through a combination of selective breeding and addition of mineral-rich fertilisers offer a practical solution to address micronutrient deficiencies such as zinc, selenium, and iron, and are particularly valuable given the high costs and logistical challenges associated with micronutrient supplementation [15,16]. Also, large-scale food fortification efforts aimed at enhancing micronutrient intake through the addition of fortificants during food processing face significant practical and logistical limitations that reduce their reach and affordability in some LMICs. For example, in Pakistan, 40–60% of flour is sourced from non-industrial mills [17], meaning that the impact of flour fortification through large industrial mills is already limited. This, alongside the need for infrastructure and quality assurance monitoring, means that these programs have failed to achieve the necessary reach and scalability [18]. Targeted biofortification of traditional plants through conventional breeding and mineral fertilisation, either alone or synergistically, appears promising in such settings. As this strategy does not disrupt the usual food preparation and consumption behaviours, it has the potential for sustainable scale-up and accessibility to vulnerable groups [4,16]. Studies have shown that consuming biofortified cereals can increase daily zinc intake by 21% to 169% compared to standard varieties [4,19,20,21,22,23,24,25]. However, there is a paucity of evidence on whether this increase translates into improved human health outcomes. The BiZiFED program, launched in 2017, is a large-scale research initiative aimed at assessing the potential of biofortified wheat to address zinc deficiency at the population level in Pakistan. This program used the high-zinc wheat variety “Zincol-2016”, introduced in 2016 in Pakistan, and evaluated its effectiveness, feasibility, and acceptability within local communities. The program included an efficacy trial (BiZiFED1, 2017–2019) [26] and an effectiveness trial (BiZiFED2, 2019–2021) [27], focusing on health outcomes in women, adolescent girls, and children living in the Khyber Pakhtunkhwa province of north-west Pakistan. The Zincol-2016 grain and flour demonstrated significantly higher zinc content than the standard non-biofortified control, leading to increased zinc intake in the participants [14,23]. The primary focus of the BiZiFED2 trial was to evaluate the impact of zinc-biofortified wheat consumption on biomarkers of zinc status, growth, and zinc-related morbidity in adolescent girls, recognising their high physiological zinc requirements and the public health relevance of addressing the intergenerational cycle of malnutrition. As biofortified wheat was provided to the entire household in this study, children were included as a secondary population to assess potential spillover effects on their growth and morbidity, considering zinc deficiency also adversely affects children in multiple ways, as previously discussed. This paper presents the findings on growth and zinc-related morbidity, which are registered as secondary outcomes, in adolescent girls (10–16 years) and children (1–5 years) from the BiZiFED2 trial. These outcomes were analysed in response to consuming flour milled from Zincol-2016 wheat, grown with zinc fertilisers and referred to as “biofortified” wheat in this paper.

## 2. Materials and Methods

### 2.1. Study Setting and Design

The current analyses are based on secondary outcomes of the Biofortified Zinc Flour to Eliminate Deficiency study (BiZiFED2). This was a large, double-blind, cluster-randomised, controlled trial conducted in two neighbouring catchment areas of rural Khyber Pakhtunkhwa, Pakistan, between November 2019 and March 2021. Ethical approval for this study was obtained from the University of Central Lancashire STEMH Ethics Committee (reference: STEMH 1014) and the Khyber Medical University Ethics Committee (reference: DIR/KMU-EB/BZ/000683). The trial is registered with the ISRCTN registry (registration number: ISRCTN17107812). The BiZiFED2 randomised controlled trial (RCT) was designed and conducted in accordance with the Consolidated Standards of Reporting Trials (CONSORT) guidelines for cluster-randomised trials, as outlined in the previous publication [14]. The complete protocol, encompassing the study setting, sample size estimation, participant recruitment and consent procedures, data monitoring for adverse events, and intervention randomisation methods, has been thoroughly detailed elsewhere [27]. Deviations to the original protocol due to the COVID-19 pandemic, along with findings on the effect of the intervention on haematological biomarkers in adolescent girls, including plasma zinc concentration (PZC), which was the primary outcome of this study, have been published subsequently [14].

Briefly, the eligibility criteria required households to have at least one unmarried, non-pregnant, non-lactating adolescent girl (10–16 years) and one child (1–5 years). In June 2019, two catchment areas located 30–40 km south-east of Peshawar, comprising a total of 44 clusters, were assessed for eligibility. A cluster-randomised design was employed in the BiZiFED2 RCT to minimise contamination between intervention and control groups, acknowledging the common practice of food sharing among neighbouring households in the study communities. Household size guided the cluster selection process, starting with smaller households and continuing until the target of 500 adolescent–child pairs was reached. This sample size was determined to adequately power the trial for the primary outcome measure, PZC for adolescent girls, and was achieved after successfully recruiting 486 eligible households, arranged in 34 clusters.

Consent was initially sought from the household head, in accordance with cultural norms, after explaining the purpose of this study. Upon obtaining their consent, the adolescent girl and the mother of the child were individually approached, and participant information sheets were provided in the local language (Pushto), along with verbal explanations to ensure its comprehension. Consent was documented either by initials or by marking an “X” for participants who were unable to write. Clusters were matched by household size and age of the participating adolescent girls to ensure comparability, and random allocation to the intervention (biofortified flour) or control (standard flour) was conducted using a computer-generated software, with each cluster having an equal probability of receiving either biofortified or control flour. The randomisation allocation was concealed from field team members, participants, and data analysts, with only the study director aware of the allocation. Detailed procedures for masking during storage, milling, and distribution are provided elsewhere [14].

Households initially received standard wheat flour for 10.5 months, which was an equilibration phase, at the end of which (August–September 2020) anthropometric measurement (adolescent girls and children) and spirometry (adolescent girls only) were performed along with other biological sample and data collection. This was followed by a 25-week duration of the intervention phase concluding in mid-March 2021, where all participating households received either the control flour (standard Galaxy variety) or the biofortified flour (Zincol-2016), according to their assigned study arm. This duration for the intervention was considered based on feasibility and the expected dietary exposure required to observe potential changes in functional outcomes. The zinc content of the biofortified and control flour was 20.7 ± 5.6 and 17.0 ± 2.6 mg/kg, respectively [14]. With a daily flour intake of 405 g, the biofortified flour contributed an estimated additional 1.5 mg of zinc per day for adolescent girls [14]. The anthropometric measurements and spirometry test were repeated at midpoint and endline of the intervention phase. Data on morbidity were collected fortnightly throughout this period, with a total of 14 rounds during the intervention period.

### 2.2. Field Procedures

At enrolment, an interviewer-administered questionnaire was used to collect data on the participant characteristics, including age, education level achieved for adolescent girls, whether attending school, and gender for children, as well as household-level data such as demographics, indicators of socioeconomic status, and living conditions (e.g., water source, kitchen, and toilet facilities).

Anthropometric measurements, including height, weight, and mid-upper arm circumference (MUAC), were taken from adolescent girls and children at the beginning, middle, and end of the intervention period by trained staff. Additional measurements included waist and hip circumference in adolescent girls and head circumference (HC) in children at the three time points. For children less than two years of age, recumbent length was assessed instead of height.

Weight was measured with the participants without shoes or heavy clothing, using a standard calibrated digital scale (Camry, Kowloon, Hong Kong), to the nearest 100 g. Recumbent length was taken using infantometer (Seca, Hamburg, Germany) to the nearest 0.1 cm by positioning with the body flat and the midline centred on the measuring board. Height was measured to the nearest 0.1 cm using a portable stadiometer (Seca, Leicester, UK) with participants standing barefoot with heels placed together, back of the heels, buttocks, and shoulder blades touching the back plate, and the head placed in the Frankfort horizontal plane. All the circumference-related measurements were taken using non-stretchable measuring tape (ABN^®^, Padalarang, Indonesia) to the nearest 0.1 cm. MUAC was measured at a point equidistant between the acromion process of the left scapula and the olecranon process of the left ulna. HC was determined by measuring the maximum occipitofrontal circumference. Waist circumference was measured at the midpoint between the lower rib and the upper margin of the iliac crest and hip circumference was measured at the maximum protuberance of the buttocks.

Body mass index (BMI) was calculated from weight (kg)/height (m)^2^. Z-scores were calculated using the World Health Organization (WHO) calculators based on the WHO Child Growth Standards [28,29]. Height-for-age Z-scores (HAZ), weight-for-age Z-scores (WAZ), and BMI-for-age Z-scores (BAZ) for participants aged above 5 years were derived using the WHO Anthroplus software version 1.0.4 [30]. For younger participants, weight-for-length/height Z-scores (WLZ/WHZ), MUAC Z-scores, and HC Z-scores (HCZ) were derived using the WHO Anthro software version 3.2.2 [31].

Stunting was defined as HAZ < −2 SD for both the population subgroups. For children, wasting was classified as WHZ < −2 SD, underweight as WAZ < −2 SD, and overweight or obesity as WHZ > +2 SD based on the WHO child growth reference. For adolescent girls, thinness was defined as a BAZ less than −2 SD, and overweight as a BAZ greater than +1 SD based on the WHO growth reference for children and adolescents aged 5–19 years. The weight-to-height ratio (WHtR) cut-off to predict abdominal or central obesity was 0.49 as determined by a recent systematic review aimed at identifying the optimal threshold of WHtR for predicting central obesity in children and adolescents [32].

Data on the incidence and duration of respiratory tract infections (RTIs) and diarrhoea in the preceding two weeks were collected fortnightly from the caregivers for children, and RTIs were self-reported by adolescents. Diarrhoea was defined as three or more loose or liquid stools per day, while RTI was characterised by symptoms such as cough, nasal discharge, or wheezing. Participants or caregivers were asked about the occurrence of specific morbidities in the past two weeks. If the response was positive, the number of days the participant experienced the morbidity was recorded. A total of 14 rounds of data was collected, with two-week intervals between consecutive rounds. The longitudinal prevalence for each morbidity was then calculated as the cumulative days of illness expressed as a percentage of the total observation days for that specific morbidity and population group.

Spirometry was performed on adolescent girls at the start, middle, and end of the intervention to assess respiratory health. The test was performed using a calibrated spirometer, Medical International Research (MIR) Spirolab^®^ (model TUK-MIR045, Rome, Italy), according to the American Thoracic Society (ATS) and European Respiratory Society (ERS) standardised guidelines [33], with participants seated and wearing a nose clip. Experienced technicians in the health centre located in the study region demonstrated the appropriate method to the participants prior to their sessions. For the present analysis, we considered forced expiratory volume in 1 s (FEV1), forced vital capacity (FVC), and the FEV1/FVC ratio as indicators of airflow limitation. These were then expressed as percent percentage predicted and Z-scores using the Global Lung Initiative (GLI) 2012 reference values with the “Other Ethnicity” equation [34,35], and the online ERS GLI calculator was employed for this purpose [36]. Z-scores with values below −1.64 of the reference (i.e., below the 5th percentile) were considered abnormal [35].

### 2.3. Statistical Analysis

All statistical analyses were undertaken using identical approaches to those adopted in our previous publication from the BiZiFED2 trial [14]. Data on the outcome variables collected at the end of the equilibration phase were treated as baseline for the intervention phase. Outcome measures relate to participants rather than clusters. The effect of intervention was assessed at the midpoint and endline of this study for all continuous variables, and comparisons between the two arms at all three time points (baseline, midpoint, and endline) were made for binary and categorical variables. All statistical analyses were conducted separately for children and adolescents to ensure that results were interpreted within their appropriate age-specific physiological contexts.

Continuous variables at the midpoint and endline were analysed using linear mixed-effects models, incorporating random cluster effects to account for the cluster-based randomisation. In addition to the study group assignments, the models included baseline values as continuous covariates. Analyses were conducted on an intention-to-treat basis, employing the restricted maximum likelihood estimation method. For the linear models, results are presented as linear regression coefficients (β) with corresponding 95% confidence intervals (CIs).

Pearson chi-squared (X^2^) tests of independence were performed at each experimental time point to conduct bivariate cross-tabulation comparisons, assessing differences in the frequency of binary or categorical outcomes (such as prevalence of stunting, underweight, etc.) between trial arms. Monte Carlo simulation was employed to calculate probability values for the chi-squared analyses. Statistical significance was set at *p* < 0.05 for all tests, with analyses conducted using the IBM SPSS software, version 29.0 (IBM Corp., Armonk, NY, USA).

## 3. Results

A total of 1034 participants, including 517 adolescent girls and 517 children, were enrolled in this study. Between enrolment and baseline data collection, 82 adolescent girls dropped out from this study, with an additional 28 dropouts occurring during the two follow-up phases (Figure 1). The reasons for withdrawals of the adolescent girls have been detailed in the CONSORT flow diagram for the BiZiFED2 RCT, along with primary outcomes in an earlier publication [14]. Each dropout of an adolescent girl necessitated the exclusion of the entire household including any children participating in this study to maintain the coherence of the study design.

### 3.1. Baseline Characteristics

The baseline characteristics from the primary outcome paper revealed the socioeconomic challenges faced by the study households [14]. Briefly, more than half (56.4%) of the adolescent girls were not attending school, with 30.3% never having attended, indicating significant educational deprivation. Economically, 61.2% of households reported a monthly income below 20,000 PKR (approximately $72), with the majority (59.1%) relying on unstable daily wage labour at brick kilns, and a mean family size of 10.5. Nearly half of the homes (46.8%) were built from mud and straw (katcha). Health data further indicated vulnerability, with 40% of households reporting diarrhoeal cases among children under five, 40% reporting RTI in this age group, and 22% reporting RTI in adolescent girls in the past month.

#### 3.1.1. Adolescent Girls

The age, educational status, and haematological parameters of the adolescent female cohort have been previously detailed [14]. At enrolment, their average age was 12.1 ± 1.7 years, nearly half (46.1%) had attained menarche, and 70% were zinc deficient based on PZC [14]. Baseline characteristics focusing on anthropometry, morbidity, and spirometry are presented in Table 1.

At baseline, the mean weight and height were 42.13 ± 10.32 kg and 148.34 ± 8.93 cm, respectively. Mean HAZ was −0.74 ± 1.12, with 10% classified as short stature (HAZ < −2). The mean BMI was 18.91 ± 3.46 kg/m^2^, with 15.9% categorised as overweight or obese and 2.8% experiencing thinness based on BMI for age Z-scores. Mean MUAC, waist circumference, and hip circumference were 22.22 ± 3.31 cm, 63.54 ± 7.98 cm, and 79.57 ± 9.06 cm, respectively. About 9.6% had a WHtR suggestive of abdominal obesity.

Spirometry data at baseline expressed as Z-scores using the GLI-2012 reference equations showed mean values of zFEV1: −0.22 ± 1.33, zFVC: −0.01 ± 1.37, and zFEV1/FVC: −0.46 ± 0.75. The proportions of participants with lung function below the 5th centile of the reference equations (lower limit of normal, LLN) were 12.3% for FEV1, 9.7% for FVC, and 6.5% for the FEV1/FVC ratio. Additionally, the longitudinal prevalence of respiratory tract infections, representing the cumulative number of days participants experienced infections as a percentage of total observed days, was 6.8% (95% CI: 5.6, 8.0) overall.

#### 3.1.2. Children

The distribution of general characteristics (such as age, gender, school attendance, and breast-feeding status) alongside functional outcomes such as growth and morbidity for children are presented in Table 2.

The mean enrolment age of children was 3.05 ± 1.03 years, and 51.6% were male and 48.4% female. Although breastfeeding practices were nearly universal, with 96.8% of children having been breastfed, only 18.3% of children were still breastfeeding at the time of enrolment. Only a small proportion of children (4.3%) were attending school at the time of enrolment.

The mean weight, height, and BMI were 14.29 ± 2.84 kg, 95.46 ± 10.25 cm, and 15.76 ± 2.16 kg/m^2^, respectively. The mean MUAC was 14.85 ± 1.08 cm and head circumference was 48.16 ± 2.24 cm. Anthropometric measures were similar in both arms of this study.

Nutritional status indicators showed that 33.8% of children were stunted (HAZ < −2), 16.5% were underweight (WAZ < −2), and 4.4% experienced wasting (WHZ < −2). Overweight or obesity (WHZ > +2) was present in 11.7% of children.

Morbidity indicators revealed the overall longitudinal prevalence of diarrhoea to be 5.5% (95% CI: 4.6, 6.4) of observed days, and respiratory tract infection was 8.9% (95% CI: 7.6, 10.1) of the observed days.

### 3.2. Impact of Intervention

#### 3.2.1. Anthropometry

##### Adolescent Girls

The treatment effect was not significant at the midpoint or endline for height, BMI, MUAC, hip circumference, HAZ, or BAZ after adjusting for baseline values (Table 3). Both weight and waist circumference increased progressively in both arms over the study period. While the control arm exhibited a slightly greater increase, the differences were not statistically significant, although a trend was observed for weight (β = −0.048; 95% CI: −0.981, 0.020; *p* = 0.059) and waist circumference (β = −0.650; 95% CI: 1.309, 0.009; *p* = 0.053) towards the end of this study (Table 3).

No significant differences were observed between the intervention and control groups for any of the indicators of malnutrition (stunting, thinness, obesity, and central obesity) either at the midpoint or endline of this study, based on the chi-squared analysis. The chi-square (X^2^) statistics for these dichotomous variables, along with their associated *p*-values and summarised data, are provided in Table 3.

##### Children

Anthropometric measurements showed increases in both trial arms over the study period. However, there were no statistically significant differences between the intervention and control arms at the midpoint or endline assessments for height, weight, MUAC, and BMI. Similarly, the derived Z-scores based on these measurements, such as WAZ, HAZ, WHZ, MUACZ, and BAZ, showed no significant differences between the arms at the two assessment points after baseline adjustments. However, a significant increase in head circumference was observed in the intervention arm compared to the control arm at endline (β = 0.432; 95% CI: 0.151, 0.713; t = 3.048; *p* = 0.003). There was also a significant intervention effect on HCZ for children under five years of age at both the midpoint (β = 0.169; 95% CI: 0.001, 0.337; t = 1.998; *p* = 0.049) and the endline assessments (β = 0.367; 95% CI: 0.149, 0.586; t = 3.342; *p* = 0.001). Linear regression coefficients (β) with 95% confidence intervals for all the above outcomes are summarised in Table 4, alongside data for anthropometric indicators for children.

Overall, a decline of close to 50% in the prevalence of underweight over the study period was observed, while the prevalence of stunting remained consistently high (33.3–40.4%). Chi-squared analyses revealed no statistically significant differences between the intervention and control arms in the prevalence of stunting at either the midpoint (X^2^_(1)_ = 1.175, *p* = 0.278) or endline (X^2^_(1)_ = 0.137, *p* = 0.711) assessments. Similarly, no significant differences were observed between arms for the prevalence of underweight at midpoint (X^2^_(1)_ = 0.241, *p* = 0.623) or endline (X^2^_(1)_ = 0.034, *p* = 0.854). Details are presented in Table 4.

#### 3.2.2. Morbidity

##### Respiratory Tract Infection

The incidence of RTIs were comparable for the two study arms until the midpoint, but towards the end of the trial some indication of a lower incidence of RTI was observed in the intervention arm for both adolescent girls (week 22: control 19.3% vs. intervention 11.5%, *p* = 0.037; week 26: control 14.5% vs. intervention 6.1%, *p* = 0.014) and children (week 26: control 27.4% vs. intervention 17.6%, *p* = 0.036). Incidence rates with corresponding chi-square and *p*-values for each round of data collection are provided in Appendix A for adolescent girls and children, respectively.

When the longitudinal prevalence of RTIs among adolescent girls and children were considered, there were no significant differences between the study arms across all time points (*p* > 0.05) after adjusting for the baseline values (Table 5). Additionally, the duration of RTIs did not show consistent meaningful differences between the study arms at data collection points falling within the midpoint or endline of this study (Appendix A). When only sick participants were considered, the average duration of illness ranged from 5 to 10 days in adolescent girls and 6 to 9 days in children for any two-week period, with no significant effect of the intervention observed at any time point (Appendix A).

##### Diarrhoea

The longitudinal prevalence of diarrhoea was consistently low across both groups and time points, with no significant differences observed either at midpoint (*p* = 0.811) or endline (*p* = 0.808). At baseline, the longitudinal prevalence was 5.8% (95% CI: 4.5, 7.2) in the control group and 5.1% (95% CI: 4.0, 6.3) in the intervention group. By the endline, the prevalence had further decreased to 3.2% (95% CI: 2.1, 4.3) in the control group and 2.9% (95% CI: 1.6, 4.2) in the intervention group (Table 5).

Incidence rates of diarrhoea showed minor variability but without statistically significant differences between the study arms across all rounds (*p* > 0.05; Appendix A). Similarly, the duration of diarrhoea was comparable between the study arms at all time points when considering all the participating children (Appendix A). When only affected children were analysed, the duration of diarrhoea ranged from 4 to 8 days for any two-week period, with no evidence of an intervention effect (Appendix A).

#### 3.2.3. Lung Function

Lung function parameters, including FEV1, FVC, and FEV1/FVC ratios, expressed as either percentage predicted or Z-scores relative to reference values, showed no significant differences between the study arms at the midpoint or endpoint, after adjusting for baseline values (Table 6). When lung function deficits were examined using Z-scores below −1.64 as a threshold (indicative of impaired lung function), the proportion of girls with abnormal lung function increased from baseline to endline irrespective of the study arms (Table 6). However, no significant differences were observed between the study arms at any assessment time points with respect to abnormal lung function (Table 6). Overall, these findings imply that the intervention had no significant impact on lung function outcomes throughout the study period.

## 4. Discussion

This study, conducted as part of the BiZiFED2 effectiveness trial, evaluated the impact of consuming the zinc-biofortified wheat variety Zincol-2016 on growth and morbidity in children and adolescent girls in rural Pakistan. In addition, lung function in the adolescent girls was also assessed.

Zinc deficiency in early life has severe and long-term consequences on growth and immune function. Stunting, a well-established population-level indicator of zinc status, remains alarmingly prevalent among children under five, affecting 40.0% in the Khyber Pakhtunkhwa province and 48.3% in Khyber Pakhtunkhwa’s newly merged districts (KP-NMD), the highest rates in the country [12]. Thus, this study provides valuable evidence on the potential health benefits of zinc biofortification in children.

While previous studies have examined the effects of zinc biofortification in children, to the best of our knowledge, this is the first trial to evaluate its effects on health and functional outcomes in adolescent girls. Adolescence represents a critical period in the life course, marked by rapid physical growth, cognitive development, and psychosocial changes. Adequate nutrition during this stage is essential, not only for immediate health benefits but also for long-term outcomes, since improving the nutritional status of adolescent girls is a preventive strategy for breaking the intergenerational cycle of malnutrition [37,38]. This focus is particularly relevant in the study region, where zinc deficiency affects nearly 70% of adolescent girls [14].

Despite modest improvements in dietary zinc intake of adolescent girls of 1.5 mg per day with biofortified flour consumption (6.9 mg/day in control vs. 8.4 mg/day in intervention), as reported in the previous publication from this trial [14], we did not find any significant effect on most of the anthropometric indicators in adolescent and children over the 25-week study period, except for a significant increase in HC and HCZ among children under five, and some indication of a reduced incidence of RTIs in both children and adolescent girls at specific time points towards the end of the trial. The duration and longitudinal prevalence of morbidities showed no consistent differences between intervention and control groups. Similarly, intake of additional zinc through biofortified wheat did not translate to positive effects on lung function parameters in adolescent girls.

Zinc deficiency is associated with impaired growth and compromised immunity [39,40,41]. Due to this well-established link between zinc with growth, stunting in children under five years of age has been proposed as a proxy indicator for identifying at-risk populations and guiding program planning for zinc interventions [42]. Supporting this association, a recent animal study demonstrated that zinc-biofortified rice significantly improved growth, as evidenced by increased body weight among zinc-deficient rats compared to those fed control diets [43]. In contrast, human studies, though limited, have provided inconsistent results, often failing to demonstrate significant beneficial effects of consuming zinc-biofortified foods on anthropometric parameters. The study by Sazawal et al. of 3000 north Indian children aged to 4–6 years evaluated the effectiveness of high-zinc biofortified wheat but did not report the anthropometric outcomes, despite these being assessed [19]. Another study in India also that reported no impact of consuming an additional 1.8–3.3 mg/day of zinc from zinc-biofortified wheat over 20 weeks on PZC or on height- or weight-based anthropometric outcomes among school-aged children [22]. Similarly, a recent study by Mehta et al. examined the effects of iron- and zinc-biofortified pearl millet, which provided an additional 1.1 mg of zinc per day for 9 months, also failed to demonstrate any significant improvements in growth indicators among children from an urban slum in India [44]. While Jongstra et al. reported no significant changes in PZC, other putative zinc biomarkers as well as anthropometric parameters following nine months of zinc-biofortified rice consumption (providing ∼1 mg/day additional zinc) in Bangladeshi preschoolers, there was a time–treatment interaction for HAZ favouring the biofortified group [25]. A plausible explanation for the differing effects could be because of the high prevalence of stunting in their cohort (59.7%), nearly double the rate observed in our study. These findings, alongside our observations on HC, highlight the necessity for further investigation into the differential effects of zinc on various growth domains.

The observed differences in HC and HCZ between the intervention and control groups in our study may be reflective of a zinc-critical role in cellular growth and brain development during early childhood. The lack of significant effects on other anthropometric indicators, such as height and weight, suggest that the benefits of zinc biofortification may be context-dependent or influenced by factors such as coexisting nutritional deficiencies and baseline status, or that indicators beyond height- and weight-based anthropometrics should be examined. Earlier studies conducted in LMICs, plus a systematic review and metanalysis investigating the effects of zinc supplementation, have demonstrated its beneficial effects on head growth in young children [45,46]. To the best of our knowledge, only one study among those testing the effectiveness of zinc-biofortified crops reported HC data. That research evaluated iron- and zinc-biofortified pearl millet in children aged 12–18 months in urban slums of Mumbai and observed no significant changes in HC [44]. This discrepancy between their finding and our study could be due to the differences in age groups studied, baseline nutritional status, and quantity and bioavailability of zinc from wheat versus pearl millet.

The intervention demonstrated some potential benefits in reducing the incidence of RTIs towards the end of this study. However, when longitudinal prevalence in our study was considered, no significant differences were observed between the intervention and control groups, indicating that the effects may have been transient or insufficient to reduce overall illness burden. A previous trial in north India reported reductions in morbidity among children consuming biofortified wheat, including a 17% reduction in days suffering from pneumonia, a 39% reduction in vomiting days, and a 9% reduction in days with fever among WRA [19]. In contrast to this, a study conducted in Bangladesh found a longitudinal prevalence ratio of 1.08 for upper respiratory tract illnesses, indicating a higher occurrence of infection among preschoolers consuming zinc-biofortified rice compared to the control group, which the authors attribute to chance [25].

The limited impact of dietary zinc interventions on diarrhoeal outcomes is consistent with findings from trials in Bangladesh and India. A study conducted with Bangladeshi children reported a longitudinal prevalence ratio for diarrhoea of 0.59 in the control group and 0.66 in the biofortified group, leading the authors to conclude that the intervention had no significant effect on diarrhoea [25]. Similarly a study in India investigating the effect of consumption of flour from high-zinc versus low-zinc biofortified wheat on morbidity among north Indian children aged 4–6 years reported an RR of 1.05 (95% CI: 0.82, 1.35) for diarrhoea [19]. A recent double-blind trial in South Indian children (aged 4–12 years; n = 273) investigated the provision of school lunches prepared with either agronomically biofortified wheat flour, postharvest-fortified wheat flour, or unfortified control wheat flour on zinc status [22]. In that study, the biofortified wheat flour and postharvest-fortified wheat flour provided an additional daily zinc intake of 1.8 and 3.3 mg over control; however, this increase in zinc intake did not significantly impact PZC. Although the study did not explicitly report outcomes related to diarrhoea or other morbidities, the authors stated that there were no significant differences in morbidity across the groups. These findings suggest that while biofortification may enhance zinc intake, its ability to improve immune function and reduce morbidity may require sustained, long-term interventions and integration with broader public health measures, possibly because of the multifactorial aetiology of infectious diseases in low-resource settings, where inadequate sanitation and persistent co-morbidities are present.

We did not find any beneficial effect of the intervention on lung function. A plausible explanation could be that lung function is influenced by a combination of genetic, environmental, and nutritional factors [47,48], and the modest increase in dietary zinc intake of 1.5 mg/day from biofortified wheat flour observed in this trial may have been insufficient to counteract these influences. It is important to note that this study was conducted in brick kiln communities, where residents live and work in environments with poor air quality. Previous studies, including a study in Northern Pakistan, found that brick kilns emit high concentrations of particulate matter and gases, contributing to poor air quality and posing significant public health risks [49,50]. Nonetheless, our study represents a novel contribution to the literature by exploring the effects of zinc biofortification on lung function in adolescent girls. Future studies should explore the interplay between zinc intakes through diet and pulmonary health over longer durations and in populations with diverse environmental exposures to derive any conclusions in this respect.

Data from this trial revealed high baseline rates of stunting (33.8% in children; 10% in adolescent girls) alongside overweight (15.9%) and central obesity (9.6%) among adolescent girls. Previously reported findings from this trial have revealed severe micronutrient deficiencies in this population, with 70% of the adolescent girls deficient in zinc and one-third exhibiting iron deficiency based on serum ferritin levels [14]. Additionally, earlier data collected in WRA in the same region demonstrated a prevalence of 30% zinc deficiency, 34% overweight, 28% obesity, and 6% underweight [13]. These findings collectively highlight the persistent challenges posed by the Double Burden of Malnutrition (DBM) in rural LMICs and emphasise the urgent need for “double duty” interventions—strategies that address both undernutrition and diet-related non-communicable diseases simultaneously [51]. Our study found that a modest increase in dietary zinc intake achieved through the consumption of zinc-biofortified wheat (Zincol-2016), did not produce statistically significant improvements in most anthropometric indicators, including weight, height, and waist circumference. However, trends toward reduced weight gain and controlled waist circumference in adolescent participants consuming biofortified flour, compared to control, suggest a potential role for zinc in modulating body composition and growth, especially in populations with a high prevalence of zinc deficiency. While these differences were not statistically significant, they highlight trends that merit further exploration, particularly with respect to overweight and obesity. At baseline, overweight and obesity prevalence was comparable between the two arms (control: 16.4%; intervention: 15.3%) and remained similar for the study duration. This suggests that biofortified wheat did not exacerbate weight gain or contribute to excess adiposity. Further to this, prior analyses from this trial showed reductions in inflammation markers such as alpha(1)-acid glycoprotein and pro-inflammatory oxylipins (5-HETE, 11-HETE, and 15-HETE), reinforcing the hypothesis that improved zinc status could attenuate inflammatory pathways linked to central obesity and metabolic dysfunction [14,52]. These trends may hint at the potential of zinc-biofortification to contribute to healthier growth trajectories and metabolic outcomes in adolescent populations. Future research should explore the biochemical mechanisms linking zinc intake via dietary interventions to anthropometric and metabolic outcomes to strengthen the evidence base for zinc biofortification as a sustainable intervention for addressing the DBM and enhancing cardiometabolic health in resource-limited settings.

This study possesses several methodological and contextual strengths: (1) a large sample size and a double-blind, randomised, cluster-controlled design; (2) focus on vulnerable subpopulations, namely young children aged 1–5 years and adolescent girls, who are at a high risk of zinc deficiency; (3) emphasis on biofortified wheat that represents a sustainable and scalable approach to addressing micronutrient deficiencies in resource-limited settings; (4) high compliance rates [14]; and (5) inclusion of comprehensive health outcomes, anthropometry, morbidity, and lung function, in addition to previously published haematological assessments, to provide a multidimensional assessment of the potential impact of the intervention.

Despite these strengths, several limitations must be acknowledged. Our study was powered to detect changes in PZC, the primary outcome, while secondary outcomes, such as anthropometric indicators and morbidity, may have been underpowered, thereby limiting the ability to detect meaningful differences. Furthermore, dietary zinc intake among children was not recorded, and blood samples were not collected, restricting the capacity to correlate actual zinc consumption and haematological zinc status with health outcomes. The relatively modest zinc differential (3.7 mg/kg) between the intervention and control flour, due to the higher-than-anticipated zinc content in the control flour [14], could have likely diminished the potential effect size. This additional zinc intake only lowered the phytate/zinc molar ratio to 18.7 in the biofortified flour, compared to 22.8 in the control flour, both of which remain above the threshold of 15, indicating low zinc bioavailability. However, the modest increase in zinc intake should not be considered solely a limitation, as it reflects the real-world conditions central to the “effectiveness” focused design of our study. Additionally, the intervention duration of 25 weeks may have been insufficient to capture significant changes in linear growth and other long-term health parameters. Finally, the findings are context-specific to rural north-west Pakistan and should be generalised with caution to similar settings with comparable sociodemographic and environmental conditions.

## 5. Conclusions

This study assessed the impact of zinc-biofortified wheat consumption on growth and morbidity in young children (1–5 years) and adolescent girls (10–16 years) in rural Pakistan, as well as on lung function in adolescent girls. While the intervention demonstrated significant improvements in head circumference and HCZ among children, it did not significantly affect most growth or morbidity outcomes. These findings highlight the complexity of translating increased dietary zinc intake into functional health benefits, particularly in resource-limited settings with high baseline rates of malnutrition and environmental challenges. Future research should explore longer intervention durations and higher zinc differentials to monitor changes in functional outcomes, particularly in light of the current national strategy in Pakistan to scale-up of zinc-biofortified wheat varieties [53]. Despite some limitations, this study contributes valuable evidence to the growing body of research on zinc biofortification as a sustainable strategy to combat micronutrient deficiencies in LMICs.

## Figures and Tables

**Figure 1 nutrients-17-01137-f001:**
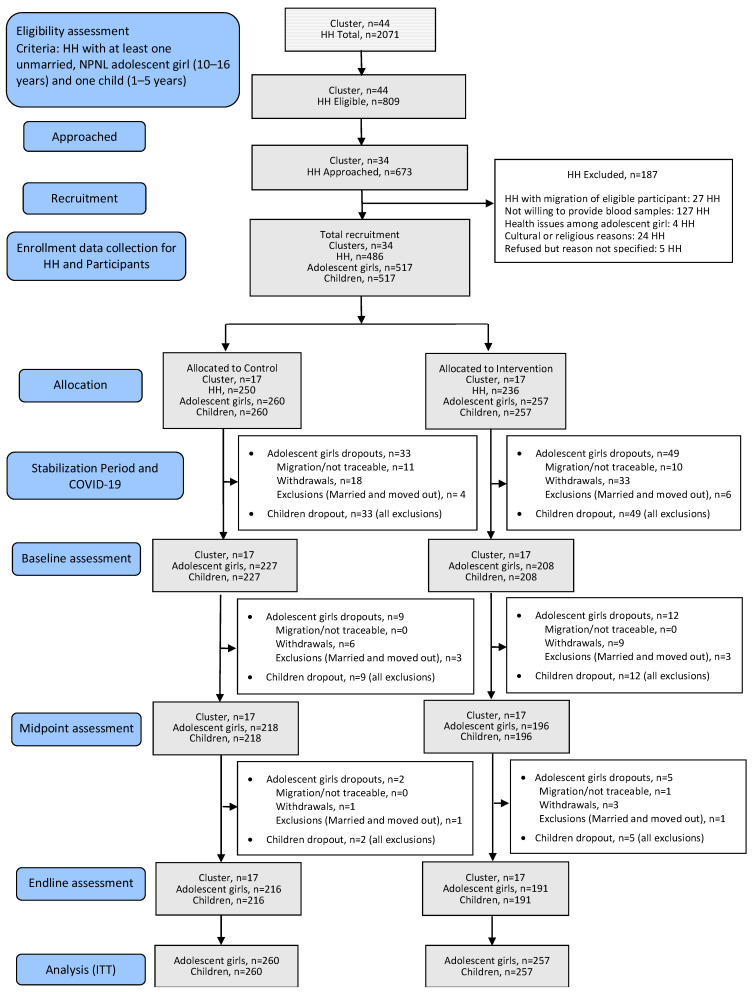
Participant flow diagram for this study. HH—household, NPNL—non-pregnant and non-lactating, ITT—intention-to-treat. Portions of this flowchart, along with detailed reasons for withdrawals, have been previously published [14].

**Table 1 nutrients-17-01137-t001:** Baseline values for anthropometry, lung function, and morbidity outcome measures of the enrolled adolescent girls.

	n	Overall	n	Control	n	Intervention
**General**						
Age (years)	517	12.11 ± 1.71	260	12.16 ± 1.69	257	12.07 ± 1.73
**Anthropometry**						
Weight (kg)	433	42.13 ± 10.32	227	42.71 ± 9.83	206	41.49 ± 10.82
Height (cm)	430	148.34 ± 8.93	226	149.26 ± 8.75	204	147.31 ± 9.04
HAZ	430	−0.74 ± 1.12	226	−0.62 ± 1.19	204	−0.86 ± 1.02
BMI (kg/m^2^)	429	18.91 ± 3.46	226	18.96 ± 3.32	203	18.85 ± 3.61
BAZ	429	−0.09 ± 1.11	226	−0.06 ± 1.09	203	−0.12 ± 1.14
MUAC (cm)	433	22.22 ± 3.31	227	22.30 ± 3.18	206	22.13 ± 3.46
Waist circumference (cm)	431	63.54 ± 7.98	226	63.53 ± 7.49	205	63.55 ± 8.50
Hip circumference (cm)	430	79.57 ± 9.06	226	80.19 ± 9.08	204	78.89 ± 9.02
WHR	427	0.80 ± 0.06	225	0.79 ± 0.05	202	0.81 ± 0.06
WHtR	427	0.43 ± 0.05	225	0.43 ± 0.04	202	0.43 ± 0.05
Stunted, HAZ < −2	430	43 (10.0)	226	21 (9.3)	204	22 (10.8)
Thinness, BAZ < −2	429	12 (2.8)	226	7 (3.1)	203	5 (2.5)
Overweight or obesity, BAZ > 1	429	68 (15.9)	226	37 (16.4)	203	31 (15.3)
Central obesity, WHtR > 0.49	427	41 (9.6)	225	16 (7.1)	202	25 (12.4)
Lung function						
FEV1 (% Pred, GLI-O)	414	97.21 ± 15.75	220	97.94 ± 16.49	194	96.39 ± 14.85
zFEV1	414	−0.22 ± 1.33	220	−0.16 ± 1.39	194	−0.29 ± 1.25
FVC (% Pred, GLI-O)	414	100.04 ± 15.49	220	101.07 ± 15.95	194	98.87 ± 14.91
zFVC	414	−0.01 ± 1.37	220	0.07 ± 1.41	194	−0.12 ± 1.32
FEV1/FVC (% Pred, GLI-O)	414	96.87 ± 5.39	220	96.52 ± 5.43	194	97.26 ± 5.33
zFEV1/FVC	414	−0.46 ± 0.75	220	−0.51 ± 0.74	194	−0.39 ± 0.76
FEV1 < −1.64 Z-score, GLI-O	414	51 (12.3)	220	24 (10.9)	194	27 (13.9)
FVC < −1.64 Z-score, GLI-O	414	40 (9.7)	220	17 (7.7)	194	23 (11.9)
FEV1/FVC < −1.64 Z-score, GLI-O	414	27 (6.5)	220	16 (7.3)	194	11 (5.7)
**Morbidity**						
Longitudinal prevalence ^#^, RTI	434	6.8 (5.6, 8.0)	226	8.2 (6.24, 10.26)	208	5.19 (3.90, 6.48)

Data presented as mean ± SD, mean (95% CI), or n (%). HAZ, height-for-age Z-score; BMI, body mass index; BAZ, BMI-for-age Z-score; MUAC, mid-upper arm circumference; WHR, waist-to-hip ratio; WHtR, weight-to-height ratio; FEV1, forced expiratory volume in 1 s; FVC, forced vital capacity; % Pred, percentage predicted; GLI-O, according to Global Lung Function Initiative—“Other ethnicity” reference; zFEV1, FEV1 Z-score based on GLI-O; zFVC, FVC Z-score based on GLI-O; zFEV1/FVC, FEV1-to-FVC Z-score based on GLI-O; RTI, respiratory tract infection. All parameters, with the exception of age, were measured at the beginning of the intervention phase. Age is reported as recorded at the time of enrolment. ^#^ Longitudinal prevalence defined as cumulative number of days participants experienced infections as a percentage of total observed days.

**Table 2 nutrients-17-01137-t002:** Baseline characteristics of the enrolled children.

	n	Total	n	Control	n	Intervention
**General**						
Age (years)	517	3.05 ± 1.03	260	3.04 ± 1.02	257	3.06 ± 1.03
Gender	517		260		257	
Male		267 (51.6)		125 (48.1)		142 (55.3)
Female		250 (48.4)		135 (51.9)		115 (44.7)
Ever breastfed	505		254		251	
Yes		489 (96.8)		247 (97.2)		242 (96.4)
No		16 (3.2)		7 (2.8)		9 (3.6)
Currently breastfeeding	480		239		241	
Yes		88 (18.3)		46 (19.2)		42 (17.4)
No		392 (81.7)		193 (80.8)		199 (82.6)
Attending school	509		256		253	
Yes		22 (4.3)		10 (3.9)		12 (4.7)
No		487 (95.7)		246 (96.1)		241 (95.3)
**Anthropometry**						
Weight (kg)	425	14.29 ± 2.84	220	14.25 ± 2.94	205	14.33 ± 2.73
Height (cm)	414	95.46 ± 10.25	216	95.58 ± 10.46	198	95.34 ± 10.01
BMI (kg/m^2^)	407	15.76 ± 2.16	211	15.74 ± 2.28	196	15.79 ± 2.04
MUAC (cm)	430	14.85 ± 1.08	225	14.86 ± 1.04	205	14.85 ± 1.12
HC (cm)	416	48.16 ± 2.24	220	48.12 ± 2.20	196	48.20 ± 2.29
WHZ	354	0.07 ± 1.46	188	0.00 ± 1.53	166	0.15 ± 1.37
HAZ	414	−1.28 ± 1.67	216	−1.19 ± 1.76	198	−1.39 ± 1.57
WAZ	425	−0.77 ± 1.26	220	−0.76 ± 1.33	205	−0.80 ± 1.19
BAZ	407	0.13 ± 1.46	211	0.09 ± 1.53	196	0.16 ± 1.37
MUACZ	381	−0.93 ± 0.82	205	−0.91 ± 0.82	176	−0.95 ± 0.82
HCZ	370	−0.95 ± 1.37	200	−0.95 ± 1.35	170	−0.94 ± 1.39
Stunting, HAZ < −2	414	140 (33.8)	216	74 (34.3)	198	66 (33.3)
Underweight, WAZ<−2	425	70 (16.5)	220	37 (16.8)	205	33 (16.1)
Wasting, WHZ < −2 SD	427	19 (4.4)	214	12 (5.6)	213	7 (3.3)
Overweight or obesity, WHZ > 2	427	50 (11.7)	214	25 (11.7)	213	25 (11.7)
**Morbidity**						
Longitudinal prevalence ^#^, Diarrhoea	434	5.5 (4.6, 6.4)	226	5.8 (4.5, 7.2)	208	5.1 (4.0, 6.3)
Longitudinal prevalence ^#^, RTI	434	8.9 (7.6, 10.1)	226	8.6 (6.9, 10.4)	208	9.1 (7.4, 10.9)

Data presented as mean ± SD, mean (95% CI), or n (%). BMI, body mass index; MUAC, mid-upper arm circumference; HC, head circumference; WHZ, weight-for-height Z-score; HAZ, height-for-age Z-score; WAZ, weight-for-age Z-score; BAZ, BMI-for-age Z-score; MUACZ, mid-upper arm circumference-for-age Z-score; HCZ, head circumference-for-age Z-score; RTI, respiratory tract infection. All parameters, with the exception of age, were measured at the beginning of the intervention phase. Age is reported as recorded at the time of enrolment. ^#^ Longitudinal prevalence defined as cumulative number of days participants experienced infections as a percentage of total observed days.

**Table 3 nutrients-17-01137-t003:** Anthropometric status of adolescent girls over time by study arms.

	Time Points	n	Control	n	Intervention	β (95% CI) *	X^2^	t	*p*
Weight (kg)	Baseline	227	42.71 ± 9.83	206	41.49 ± 10.82				
	Midpoint	217	43.85 ± 9.84	194	42.39 ± 10.55	−0.191 (−0.548, 0.167)		−1.086	0.285
	Endline	215	45.47 ± 9.73	191	43.43 ± 10.15	−0.0480 (−0.981, 0.020)		−1.952	0.059
Height (cm)	Baseline	226	149.26 ± 8.75	204	147.31 ± 9.04				
	Midpoint	216	149.81 ± 8.68	191	148.00 ± 8.60	−0.028 (−0.244, 0.189)		−0.259	0.796
	Endline	213	150.56 ± 8.52	189	148.66 ± 8.45	−0.065 (−0.449, 0.319)		−0.339	0.736
BMI (kg/m^2^)	Baseline	226	18.96 ± 3.32	203	18.85 ± 3.61				
	Midpoint	216	19.39 ± 3.35	191	19.08 ± 3.48	−0.064 (−0.237, 0.108)		−0.759	0.453
	Endline	213	19.93 ± 3.33	189	19.42 ± 3.32	−0.216 (−0.501, 0.068)		−1.537	0.132
MUAC (cm)	Baseline	227	22.30 ± 3.18	206	22.13 ± 3.46				
	Midpoint	216	22.65 ± 3.23	190	22.31 ± 3.42	−0.023 (−0.175, 0.129)		−0.312	0.757
	Endline	214	23.14 ± 3.31	187	22.76 ± 3.30	−0.005 (−0.259, 0.249)		−0.040	0.968
Waist circumference (cm)	Baseline	226	63.53 ± 7.49	205	63.55 ± 8.50				
	Midpoint	216	64.36 ± 7.39	191	63.92 ± 8.47	−0.236 (0.622, 0.150)		−1.239	0.223
	Endline	209	65.59 ± 7.52	187	64.85 ± 8.35	−0.650 (1.309, 0.009)		−1.989	0.053
Hip circumference (cm)	Baseline	226	80.19 ± 9.08	204	78.89 ± 9.02				
	Midpoint	217	81.08 ± 9.00	189	79.51 ± 9.05	0.052 (−0.392, −0.495)		0.234	0.816
	Endline	215	82.35 ± 8.86	186	80.44 ± 8.69	−0.301 (−0.882, 0.281)		−1.044	0.302
BAZ	Baseline	226	−0.06 ± 1.09	203	−0.12 ± 1.14				
	Midpoint	216	0.03 ± 1.08	191	−0.08 ±1.11	−0.013 (−0.086, 0.061)		−0.353	0.726
	Endline	213	0.19 ± 1.00	189	0.02 ± 1.05	−0.067 (−0.181, 0.047)		−1.196	0.239
HAZ	Baseline	226	−0.62 ± 1.19	204	−0.86 ± 1.02				
	Midpoint	216	−0.71 ± 1.16	191	−0.90 ± 0.98	0.001 (−0.028, 0.031)		0.093	0.926
	Endline	213	−0.73 ± 1.14	189	−0.92 ± 1.01	0.006 (−0.046, 0.059)		0.247	0.806
WHtR	Baseline	225	0.43 ± 0.04	202	0.43 ± 0.05				
	Midpoint	215	0.43 ± 0.05	188	0.43 ± 0.05	−0.002 (−0.004, 0.001)		−1.220	0.232
	Endline	207	0.44 ± 0.05	185	0.44 ± 0.05	−0.004 (−0.009, 0.001)		−1.703	0.960
Stunting, HAZ < −2	Baseline	226	21 (9.3)	204	22 (10.8)		0.265		0.607
	Midpoint	216	20 (9.3)	191	21 (11.0)		0.337		0.562
	Endline	213	23 (10.8)	189	22 (11.6)		0.071		0.789
Thinness, BAZ < −2	Baseline	226	7 (3.1)	203	5 (2.5)		0.158		0.691
	Midpoint	216	7 (3.2)	191	8 (4.2)		0.256		0.613
	Endline	213	2 (0.9)	189	5 (2.6)		1.705		0.192
Overweight or obese, BAZ > 1	Baseline	226	37 (16.4)	203	31 (15.3)		0.097		0.755
	Midpoint	216	41 (19.0)	191	30 (15.7)		0.755		0.385
	Endline	213	45 (21.1)	189	31 (16.4)		1.458		0.227
Central obesity, WHtR > 0.49	Baseline	225	16 (7.1)	202	25 (12.4)		3.399		0.065
	Midpoint	215	16 (7.4)	188	23 (12.2)		2.635		0.105
	Endline	207	23 (11.1)	185	24 (13.0)		0.321		0.571

Data presented as mean ± SD, mean (95% CI), or n (%). HAZ, height-for-age Z-score; BMI, body mass index; BAZ, BMI-for-age Z-score; MUAC, mid-upper arm circumference; WHtR, weight-to-height ratio. * Values represent beta coefficient and 95% CI from linear regression models. *p*-values obtained using linear mixed models adjusted for cluster effect and baseline values to test differences between the groups for continuous variables. Categorical variables tested by Pearson’s chi-squared test. Significance was set at *p* < 0.05.

**Table 4 nutrients-17-01137-t004:** Anthropometric status of children over time by study arms.

	Time Points	n	Control	n	Intervention	β (95% CI) *	X^2^	t	*p*
Weight (kg)	Baseline	220	14.25 ± 2.94	205	14.33 ± 2.73				
	Midpoint	208	15.13 ± 3.05	190	15.21 ± 2.86	0.051 (−0.193, 0.296)		0.426	0.673
	Endline	208	15.98 ± 3.09	186	15.77 ± 2.94	−0.229 (−0.570, 0.111)		−1.356	0.182
Height (cm)	Baseline	216	95.58 ± 10.46	198	95.34 ± 10.01				
	Midpoint	205	96.85 ± 10.59	188	96.45 ± 9.89	0.15 (−0.197, 0.497)		0.858	0.393
	Endline	198	98.65 ± 10.29	181	97.66 ± 9.87	0.396 (−0.160, 0.953)		1.412	0.161
BMI (kg/m^2^)	Baseline	211	15.74 ± 2.28	196	15.79 ± 2.04				
	Midpoint	198	16.26 ± 2.55	182	16.29 ± 2.04	1.593 (−0.253, 0.437)		0.502	0.618
	Endline	192	16.44 ± 2.50	176	16.37 ± 2.10	−0.183 (−0.693, 0.326)		−0.720	0.475
MUAC (cm)	Baseline	225	14.86 ± 1.04	205	14.85 ± 1.12				
	Midpoint	213	15.18 ± 1.12	194	15.25 ± 1.12	0.072 (−0.086, 0.231)		0.919	0.362
	Endline	214	15.56 ± 1.13	189	15.59 ± 1.18	0.057 (−0.168, 0.282)		0.510	0.612
HC (cm)	Baseline	220	48.12 ± 2.20	196	48.20 ± 2.29				
	Midpoint	214	48.26 ± 2.13	194	48.58 ± 1.92	0.188 (−0.038, 0.414)		1.658	0.101
	Endline	213	48.47 ± 2.03	190	48.76 ± 1.82	0.432 (0.151, 0.713)		3.048	0.003
WHZ	Baseline	188	0.00 ± 1.53	166	0.15 ± 1.37				
	Midpoint	165	0.46 ± 1.65	146	0.51 ± 1.33	0.067 (−0.174, 0.308)		0.558	0.580
	Endline	147	0.61 ± 1.62	130	0.55 ± 1.35	−0.121 (−0.469, 228)		−0.695	0.490
HAZ	Baseline	216	−1.19 ± 1.76	198	−1.39 ± 1.57				
	Midpoint	205	−1.33 ± 1.76	188	−1.53 ± 1.54	0.022 (−0.072, 0.116)		0.474	0.637
	Endline	198	−1.28 ± 1.72	181	−1.57 ± 1.53	0.091 (−0.057, 0.240)		1.221	0.225
WAZ	Baseline	220	−0.76 ± 1.33	205	−0.80 ± 1.19				
	Midpoint	208	−0.56 ± 1.34	190	−0.59 ± 1.17	0.019 (−0.109, 0.148)		0.306	0.761
	Endline	208	−0.38 ± 1.25	186	−0.53 ± 1.14	−0.105 (−0.278, 0.068)		−1.228	0.226
BAZ	Baseline	211	0.09 ± 1.53	196	0.16 ± 1.37				
	Midpoint	198	0.46 ± 1.65	182	0.54 ± 1.32	0.104 (−0.124, 0.332)		0.920	0.362
	Endline	192	0.60 ± 1.55	176	0.61 ± 1.34	−0.022 (−0.337, 0.293)		−0.141	0.889
HCZ	Baseline	200	−0.95 ± 1.35	170	−0.94 ± 1.39				
	Midpoint	183	−0.99 ± 1.27	159	−0.78 ± 1.19	0.169 (0.001, 0.337)		1.998	0.049
	Endline	162	−0.91 ± 1.12	141	−0.73 ± 1.08	0.367 (0.149, 0.586)		3.342	0.001
MUACZ	Baseline	205	−0.91 ± 0.82	176	−0.95 ± 0.82				
	Midpoint	182	−0.72 ± 0.85	159	−0.73 ± 0.78	0.067 (−0.066, 0.201)		1.007	0.318
	Endline	163	−0.49 ± 0.84	140	−0.52 ± 0.83	0.079 (−0.121, 0.278)		0.792	0.432
Stunting, HAZ < −2	Baseline	216	74 (34.3)	198	66 (33.3)		0.040		0.842
	Midpoint	205	72 (35.1)	188	76 (40.4)		1.175		0.278
	Endline	198	74 (37.4)	181	71 (39.2)		0.137		0.711
Underweight, WAZ < −2	Baseline	220	37 (16.8)	205	33 (16.1)		0.040		0.841
	Midpoint	208	24 (11.5%)	190	25 (13.2)		0.241		0.623
	Endline	208	19 (9.1)	186	18 (9.7)		0.034		0.854

Data presented as mean ± SD, mean (95% CI), or n (%). BMI, body mass index; MUAC, mid-upper arm circumference; HC, head circumference. WHZ, weight-for-height Z-score; HAZ, height-for-age Z-score; WAZ, weight-for-age Z-score; BAZ, BMI-for-age Z-score; MUACZ, mid-upper arm circumference-for-age Z-score; HCZ, head circumference-for-age Z-score. * Values represent beta coefficient and 95% CI from linear regression models. *p*-values obtained using linear mixed models adjusted for cluster effect and baseline values to test differences between the groups for continuous variables. Categorical variables tested by Pearson’s chi-squared test. Significance was set at *p* < 0.05.

**Table 5 nutrients-17-01137-t005:** Longitudinal prevalence of morbidities among participants by study arm at baseline, midpoint, and endline.

	n	Control	n	Intervention	β (95% CI) *	t	*p*
RTI in adolescent girls							
Baseline	226	8.2 (6.2, 10.3)	208	5.2 (3.9, 6.5)			
Midpoint	218	10.6 (8.6, 12.7)	196	9.9 (8.0, 11.9)	−0.608 (−5.124, 3.909)	−0.270	0.788
Endline	215	8.1 (6.4, 9.8)	191	6.3 (4.9, 7.7)	−2.098 (−5.851, 1.656)	−1.121	0.267
RTI in children							
Baseline	226	8.6 (6.9, 10.4)	208	9.1 (7.4, 10.9)			
Midpoint	218	16.5 (14.1, 18.9)	196	14.8 (12.5, 17.2)	−3.425 (−9.143, 2.293)	−1.194	0.237
Endline	215	14.0 (11.5, 16.4)	191	12.2 (9.9, 14.5)	−3.28 (−9.66, 3.100)	−1.020	0.310
Diarrhoea in children							
Baseline	226	5.8 (4.5, 7.2)	208	5.1 (4.0, 6.3)			
Midpoint	218	4.7 (3.2, 6.1)	196	4.9 (3.5, 6.2)	0.331 (−2.449, 3.110)	0.240	0.811
Endline	215	3.2 (2.1, 4.3)	191	2.9 (1.6, 4.2)	−0.255 (−2.374, 1.865)	−0.245	0.808

Values represent mean (95% CI). RTI—respiratory tract infection. * Values represent beta coefficient and 95% CI from linear regression models. *p*-values obtained using linear mixed models adjusted for cluster effect and baseline values to test differences between the groups for continuous variables. Significance was set at *p* < 0.05. Morbidity data were collected every two weeks covering a total period of 28 weeks through 14 rounds of data collection. Data collected in rounds 1–4 were considered as the baseline period, rounds 5–9 as the midpoint, and rounds 10–14 as the endline period.

**Table 6 nutrients-17-01137-t006:** Lung function test of adolescent female participants by study arm at baseline, midpoint, and endline.

	n	Control	n	Intervention	β (95% CI) *	X^2^	t	*p*
FEV1 (% Pred, GLI-O)								
Baseline	220	97.94 ± 16.49	194	96.39 ± 14.85				
Midpoint	214	92.62 ± 14.62	187	92.56 ± 14.10	0.733 (−1.766, 3.232)		0.577	0.564
Endline	212	90.89 ± 14.34	187	90.46 ± 13.83	−0.097 (−2.694, 2.499)		−0.074	0.941
zFEV1								
Baseline	220	−0.16 ± 1.39	194	−0.29 ± 1.25				
Midpoint	214	−0.61 ± 1.23	187	−0.61 ± 1.18	0.062 (−0.148, 0.271)		0.581	0.562
Endline	212	−0.75 ± 1.20	187	−0.79 ± 1.15	−0.008 (−0.225, 0.209)		−0.074	0.941
FVC (% Pred, GLI-O)								
Baseline	220	101.07 ± 15.95	194	98.87 ± 14.91				
Midpoint	214	97.61 ± 14.84	187	97.93 ± 13.69	1.240 (−1.588, 4.068)		0.910	0.373
Endline	212	95.21 ± 14.17	187	94.74 ± 14.05	0.042 (−2.616, 2.701)		0.031	0.975
zFVC								
Baseline	220	0.07 ± 1.41	194	−0.12 ± 1.32				
Midpoint	214	−0.23 ± 1.32	187	−0.20 ± 1.22	0.111 (−0.140, 0.361)		0.916	0.370
Endline	212	−0.44 ± 1.25	187	−0.48 ± 1.25	0.002 (−0.234, 0.237)		0.015	0.988
FEV1/FVC (% Pred, GLI-O)								
Baseline	220	96.52 ± 5.43	194	97.26 ± 5.33				
Midpoint	214	94.71 ± 6.15	186	94.51 ± 7.34	−0.782 (−2.228, 0.664)		−1.091	0.281
Endline	212	95.29 ± 7.01	187	95.39 ± 7.12	−0.229 (−1.927, 1.468)		−0.275	0.785
zFEV1/FVC								
Baseline	220	−0.51 ± 0.74	194	−0.39 ± 0.76				
Midpoint	214	−0.77 ± 0.93	186	−0.77 ± 1.01	−0.088 (−0.296, 0.120)		−0.855	0.398
Endline	212	−0.66 ± 0.97	187	−0.64 ± 1.03	−0.030 (−0.269, 0.209)		−0.258	0.798
FEV1 <−1.64 Z-score, GLI-O								
Baseline	220	24 (10.9)	194	27 (13.9)		0.864		0.353
Midpoint	214	30 (14.0)	187	37 (19.8)		2.385		0.122
Endline	212	48 (22.6)	187	36 (19.3)		0.687		0.407
FVC <−1.64 Z-score, GLI-O								
Baseline	220	17 (7.7)	194	23 (11.9)		2.013		0.156
Midpoint	214	23 (10.7)	187	17 (9.1)		0.305		0.581
Endline	212	31 (14.6)	187	30 (16.0)		0.155		0.694
FEV1/FVC <−1.64 Z-score, GLI-O								
Baseline	220	16 (7.3)	194	11 (5.7)		0.434		0.510
Midpoint	214	32 (15.0)	186	29 (15.6)		0.031		0.859
Endline	212	23 (10.8)	187	22 (11.8)		0.083		0.773

Data presented as mean ± SD or n (%). FEV1, forced expiratory volume in 1 s; FVC, forced vital capacity; % Pred, percentage predicted; GLI-O, according to Global Lung Function Initiative—“Other ethnicity” reference; zFEV1, FEV1 Z-score based on GLI-O; zFVC, FVC Z-score based on GLI-O; zFEV1/FVC, FEV1-to-FVC Z-score based on GLI-O. * Values represent beta coefficient and 95% CI from linear regression models. *p*-values obtained using linear mixed models adjusted for cluster effect and baseline values to test differences between the groups for continuous variables. Categorical variables tested by Pearson’s chi-squared test. Significance was set at *p* < 0.05.

## Data Availability

The data will be made available at uclandata.uclan.ac.uk in due course. In the meantime, requests for access can be sent to Nicola M. Lowe.

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
