# Peer review of "The Effectiveness of Zinc-Biofortified Wheat Flour Intake on the Growth and Morbidity Outcomes of Rural Pakistani Children and Adolescent Girls: A Cluster-Randomised, Double-Blind, Controlled Trial"

_nutrients, 2025, doi:10.3390/nu17071137_

Round 1

Reviewer 1 Report

Comments and Suggestions for Authors

The study Effectiveness of zinc-bio-fortified wheat flour intake on the two growth and morbidity outcomes in rural Pakistani children and adolescent girls: a cluster-randomized, double-masked, controlled trial, by Gupta et al., evaluated a significant topic: the action of zinc on the health of adolescents and children. The action of zinc biofortified flour is one of the possible strategies for reducing the level of zinc deficiency (dietary and biochemical), nutritional short stature, stunting, and wasting. This study is part of a more extensive study, and the biochemical data has already been presented. In this paper, the intervention data refers to anthropometric data and lung function. A few things prior to analyzing the project: No plagiarism was detected. After examining the text and removing the bibliographical references, the Quetxt pro software found only 11% of plagiarism, almost always referring to self-references based on the group's previous work (https://www.mdpi.com/2072-6643/14/8/1657). Evaluating the previously published work, biofortification provided modest improvements in absorbable zinc, with little visible effect on anthropometry and zinc nutritional status, as well as on other minerals. This study highlights the robustness of the figures, the appropriate randomization, and the analyses that were carried out. However, the combination of data from adolescents and children is often confusing. The prevalence of stunting obviously occurs in children under five, while dietary zinc deficiency is more important in children than in adolescents. There is often a priority discussion for adolescents with less emphasis on children. The analyses carried out leave something to be desired, as the levels of dietary intake and zinc values for each group are not presented, which could undoubtedly modify the results presented. Dietary intake is certainly the biggest problem in analyzing the results, allowing groups with low intakes to alter the outcomes. The failure to modify hematimetry indices of iron and zinc deficiency, especially in children under five, may be related to inadequate intake of the mineral, either from fortified flour or from deficiencies in other sources, such as meat, seafood, and fortified foods. The fact that the absorption of the mineral and its bioavailability were not presented leaves room for discussion about whether the product was suitable for its purpose since, in practically all the outcomes, the result was insufficient. The paper cites its authors in numerous references, which may interfere with the analysis of the project. However, one cannot fail to recognize the importance of Gupta and Lowe in conducting public policies based on micronutrients for at-risk populations.  I want to emphasize that there does appear to be a conflict of interest due to the participation of suppliers and companies, which should be mentioned under conflicts of interest. In summary, the discussion and analysis are almost always carried out with two different groups in terms of intake, anthropometry, risk of infection, environment, and type of diet at baseline, indicating that there is possible statistical interference and data analysis. The paper is exceptionally well-written and may indicate a direction in the analysis of intervention projects with biofortified foods. We miss articles from other regions with papers on the subject, highlighting the intervention of biofortified flours, milk, meat, and other cereals (see Embrapa's work in Brazil).  

Reviewer 2 Report

Comments and Suggestions for Authors

This is an interesting clinical trial with adequate novelty. However, some points should be addressed.

  • The gender of the enrolled children should be reported in both the Abstract and the following analysis of the data of the study.
  • Zinc bioavailability reported in line 51 should be more deeply discussed.
  • The sentence in line 56 "Zinc is associated with respiratory health and diarrhea in vulnerable populations" should also more deeply discussed as this is the central topic of the present study.
  • The sentence in lines 78-82 is too long and not easily readable. It could be split into two smaller sentences.
  • The authors should provide a justification why they applied a 25-week duration of intervention phase.
  • The resolution and the quality of Figure 1 should be improved.
  • A separate analysis could be performed for females children and males children since the adolescents participants were only girls.
  • A justification for the analysis of both meales and females children compared to the analysis of girls adolescents should be reported.
  • A justification should also be added concerning the reasons that the authors included onle girls adolescents in their study.
  • The 3rd paragraph of the Discussion section is too long and it included some repetitions. It could be split into two smaller paragraphs by reducing a bit their size to avoid repetitions with the results of the study.

Round 2

Reviewer 2 Report

Comments and Suggestions for Authors

The authors have significantly improved their manuscript.